# Estimating the global demand curve for a leishmaniasis vaccine: A generalisable approach based on global burden of disease estimates

**Sakshi Mohan**[1]*, **Paul Revill**[1], **Stefano Malvolti**[2], **Melissa Malhame**[2], **Mark Sculpher**[1], **Paul M. Kaye**[3]

**1** Center for Health Economics (CHE), University of York, York, United Kingdom, **2** MMGH Consulting, Zurich, Switzerland, **3** York Biomedical Research Institute, Hull York Medical School, University of York, York, United Kingdom

* sakshi.mohan@york.ac.uk

**Data Availability Statement:** The primary data are within the manuscript and its Supporting Information files. The excel tool which performs

## Abstract

### Background

A pressing need exists to develop vaccines for neglected diseases, including leishmaniasis. However, the development of new vaccines is dependent on their value to two key players–vaccine developers and manufacturers who need to have confidence in the global demand in order to commit to research and production; and governments (or other international funders) who need to signal demand based on the potential public health benefits of the vaccine in their local context, as well as its affordability. A detailed global epidemiological analysis is rarely available before a vaccine enters a market due to lack of resources as well as insufficient global data necessary for such an analysis. Our study seeks to bridge this information gap by providing a generalisable approach to estimating the commercial and public health value of a vaccine in development relying primarily on publicly available Global Burden of Disease (GBD) data. This simplified approach is easily replicable and can be used to guide discussions and investments into vaccines and other health technologies where evidence constraints exist. The approach is demonstrated through the estimation of the demand curve for a future leishmaniasis vaccine.

### Methodology/Principal findings

We project the ability to pay over the period 2030–2040 for a vaccine preventing cutaneous and visceral leishmaniasis (CL / VL), using an illustrative set of countries which account for most of the global disease burden. First, based on previous work on vaccine demand projections in these countries and CL / VL GBD-reported incidence rates, we project the potential long-term impact of the vaccine on disability-adjusted life years (DALYs) averted as a result of reduced incidence. Then, we apply an economic framework to our estimates to determine vaccine affordability based on the abilities to pay of governments and global funders, leading to estimates of the demand and market size. Based on our estimates, the maximum ability-to-

the analysis (including all the underlying input data) and python code to generate the figures can be found here -https://github.com/sakshimohan/leish-vaccine.

**Funding:** SM, PR and MS were supported by UK Research and Innovation as part of the Global Challenges Research Fund, grant number MR/P028004/1. PMK was supported by a Wellcome Senior Investigator Award (Grant No. 104726) and PMK, StM and MM were supported by a Wellcome Translation Award (Grant No. 108518). The funders had no role in study design, data collection and analysis, decision to publish, or preparation of the manuscript. https://www.ukri.org/ https://wellcome.org/.

**Competing interests:** I have read the journal's policy and the authors of this manuscript have the following competing interests: PK is co-author of a patent protecting the gene insert used in Leishmania candidate vaccine ChAd63-KH (Europe 10719953.1; India 315101).

to-pay of a leishmaniasis vaccine (per course, including delivery costs), given the current estimates of incidence and population at risk, is higher than $5 for 25–30% of the countries considered, with the average value-based maximum price, weighted by quantity demanded, being $5.7–6 [$0.3 - $34.5], and total demand of over 560 million courses.

## Conclusion/Significance

Our results demonstrate that both the quantity of vaccines estimated to be required by the countries considered as well as their ability-to-pay could make a vaccine for leishmaniasis commercially attractive to potential manufacturers. The methodology used can be equally applied to other technology developments targeting health in developing countries.

### Author summary

As of 2019, between 498,000 and 862,000 new cases of all forms of leishmaniasis were estimated to occur each year resulting in up to 18,700 deaths and up to 1.6 million DALYs lost. Given low treatment coverage, poor compliance and the emergence of drug resistance, challenges in sustaining vector control strategies and the ability of parasites to persist in animal reservoirs independent of human infection, an effective vaccine could significantly reduce the health and economic burden of these diseases. However, commitment to the development of a new vaccine requires a market signal from governments and global funders who in turn require better estimates of the potential public health value of the vaccine. This study uses the development of a leishmaniasis vaccine as a case study to illustrate a generalizable approach to estimating the commercial and public health value of a technology relying primarily on publicly available GBD data. More specifically, by projecting the potential public health impact of the rollout of a leishmaniasis vaccine and translating this into monetary values based on the concept of health opportunity cost, we estimate the demand curve for such a vaccine for an 11-year period between 2030 and 2040. At an estimated global demand of over 560 million courses with the average value-based maximum price, weighted by quantity demanded, of $5.7–6 [$0.3 - $34.5], our results demonstrate that both the quantity of vaccines estimated to be required by the countries considered as well as their ability-to-pay make the vaccine commercially attractive to potential manufacturers.

## Introduction

The leishmaniases represent a group of parasitic diseases, with infection to human populations transmitted by the bite of phlebotomine sand flies. Disease presentation varies because of differences in parasite and host genetics and may be influenced by additional factors such as host nutritional status or co-infection. The leishmaniases disproportionately affect populations in low- and middle-income countries (LMICs). According to the Global Burden of Disease (GBD) study 2019, between 498,000 and 862,000 new cases of all forms of leishmaniasis were estimated to occur each year resulting in up to 18,700 deaths and up to 1.6 million DALYs lost [1]. Previously designated one of the most neglected among neglected tropical diseases (NTDs) based on limited resources invested in diagnosis, treatment and control [2], leishmaniasis accounts for 4% of the global DALY burden of NTDs and 5.5% of global NTD-related

deaths. Furthermore, it is widely believed that these numbers grossly underestimate the real burden of leishmaniasis as a result of underreporting and limited understanding of the true lifetime impact of the disease [3–5].

The two most prevalent forms of leishmaniasis are localized cutaneous leishmaniasis (CL) and visceral leishmaniasis (VL). Despite the availability of effective treatment regimens, access to treatment remains low [6,7]. Given low treatment coverage, the occurrence of poor compliance and the emergence of drug resistance [8], challenges in sustaining vector control strategies [9], and the ability of parasites to persist in animal reservoirs, vaccines are widely regarded as having the potential to significantly impact the health burden posed by leishmaniasis and to contribute to regional leishmaniasis elimination campaigns [10]. Between 2007 and 2013, nearly $66 million was invested by public sector and philanthropic funders towards leishmaniasis vaccine research and development [11]. Numerous vaccine candidates have been evaluated in preclinical models of disease, but few have progressed to clinical trial stage [11]. Currently, only one therapeutic vaccine clinical trial is ongoing [12], and a genetically attenuated live *L. major* vaccine is scheduled for manufacture in 2022 and for Phase I clinical trial in 2023 [13,14].

However, it is not enough just to develop a clinically effective vaccine. Rather, the vaccine also needs to be affordable and suitable for delivery and administration in health systems. In particular, for a vaccine to be produced and used, it needs to offer value to two key players: vaccine developers and manufacturers who need to have confidence in global demand in order to commit to research and production; and governments (or other international funders) who need to be sure of the potential public health benefits of the vaccine in their local context, as well as affordability of the vaccine, in order to signal demand [14–16].

This study seeks to fill this information gap about the commercial value proposition and likely demand for a future leishmaniasis vaccine. This evaluation of a vaccine's potential economic value can also help shed light on key targets for vaccine development and manufacturing plans such as efficacy targets, target population groups/geographies, upper bound for manufacturing costs (and required scale of manufacturing), and target market size while the vaccine is under development.

More generally, this study seeks to develop a simplified and generalizable framework which employs publicly available burden of disease data to project the affordability, market size and public health value of new interventions in order to inform and spur continued product development that can improve health in low and middle-income countries (LMICs).

## Methods

### General approach

This study assesses the value associated with the introduction of a vaccine to prevent CL / VL. Value is assessed in terms of the vaccine's potential impact on mortality and morbidity taking into account its affordability within an illustrative set of countries in which the disease is endemic. First, based on previous work on vaccine demand projections in these countries [14] and CL / VL incidence rates [1], we project the potential long-term impact of a leishmaniasis vaccine on disability-adjusted life years (DALYs) averted as a result of reduced incidence. Ideally, such an analysis would require a detailed modeling of the disease epidemiology, disease dynamics, and health system capabilities of each country under consideration. However, such models are not currently available for most countries but planning for vaccine research and manufacturing needs to continue in their absence. Therefore, we sought to develop a simplified approach, which uses publicly available data on disease incidence and burden and population growth projections to assess the public health value of a future vaccine.

Second, we apply a health economic framework to our estimates of the future health impact of a vaccine to determine the vaccine's affordability based upon the abilities to pay of governments and global funders, leading to estimates of the demand and market size in this illustrative set of countries. All monetary values are presented in 2019 US Dollars (USD).

## Geographic focus

The analysis in this paper is focused on a representative sample of 24 countries belonging to a range of income levels [17], geographic regions, type of endemic leishmaniasis, and Gavi, The Vaccine Alliance (Gavi) support status [18] (**Table 1**). In 2019, these countries together represented 80% of the global DALY burden of CL and VL, and 70% and 82% of the global incidence of CL and VL respectively [1]. We had to limit our analysis to these 24 countries due to the lack of granular data on the population at risk and projected vaccine demand for other countries from Malvolti et al. (2021) [14], further described below.

## Vaccine efficacy and health effects

In the absence of a rigorous epidemiological model, we project the health effect of a vaccine using the following estimates: i) total population susceptible to the disease (or population at risk); ii) incidence of the disease among the population at risk; iii) per person burden of disease; and iv) vaccine coverage and efficacy. This sub-section describes how these estimates were obtained and used.

**Table 1. List of countries included in the analysis.**

| Country | Continent | WHO Region | World Bank Income Level | Disease endemicity | Gavi support status (2020) |
|---|---|---|---|---|---|
| Afghanistan | Asia | EMRO | Upper-middle | VL | Initial self-financing |
| Algeria | Africa | EMRO | Upper-middle | VL | Ineligible |
| Bangladesh | Asia | SEARO | Low | CL & VL | Preparatory transition |
| Brazil | South America | PAHO | Upper-middle | VL | Ineligible |
| China | Asia | WPRO | Lower-middle | CL & VL | Ineligible |
| Ethiopia | Africa | AFRO | High | CL | Initial self-financing |
| Georgia | Europe | EURO | Lower-middle | VL | Fully self-financing |
| India | Asia | SEARO | Lower-middle | CL | Accelerated transition |
| Israel | Asia | EURO | Lower-middle | VL | Ineligible |
| Kenya | Africa | AFRO | Lower-middle | CL | Preparatory transition |
| Morocco | Africa | EMRO | Lower-middle | CL | Ineligible |
| Nepal | Asia | SEARO | Upper-middle | VL | Initial self-financing |
| Nigeria | Africa | AFRO | High | CL | Accelerated transition |
| Pakistan | Asia | EMRO | Low | VL | Preparatory transition |
| Paraguay | South America | PAHO | Low | VL | Ineligible |
| Saudi Arabia | Asia | EMRO | High | VL | Ineligible |
| Somalia | Africa | EMRO | Low | CL & VL | Initial self-financing |
| South Sudan | Africa | EMRO | Low | CL | Initial self-financing |
| Spain | Europe | EURO | Lower-middle | CL | Ineligible |
| Sudan | Africa | AFRO | Upper-middle | CL | Preparatory transition |
| Syria | Asia | EMRO | Lower-middle | CL | Initial self-financing |
| Tunisia | Africa | EMRO | Upper-middle | VL | Ineligible |
| Turkey | Asia | EURO | Upper-middle | VL | Ineligible |
| Uzbekistan | Asia | EURO | Low | CL & VL | Accelerated transition |

Environmental factors that affect the relationship between hosts, vectors (human, animal or sandfly) and the reservoir determine the risk of leishmaniasis in the population. Malvolti et al. (2021) [14] draw upon WHO Leishmaniasis country profiles as well as Pigott et al. (2014) [19] to project the size of the population at risk for leishmaniasis-endemic countries until 2040 using 5-year population growth projections from UN/DESA [20]. The age-wise composition of the population at risk was based on projection from the World Population Prospects report [21].

Incidence estimates were obtained from the Global Burden of Disease (GBD) study in 2019 [1]. These were converted into incidence rates specific to populations at risk for 2019 by dividing the incidence by the size of the population at risk (note that this assumes that no one outside the main population at risk contracts the disease) for the different age groups included in the vaccine demand projections in Malvolti et al. (2021) [14], namely 0–4 years, 5–14 years, and 15–29 years old. In the absence of epidemiological projections of leishmaniasis incidence and given that there has not been a significant decline in incidence over the last five years [22], we make the assumption that the incidence rate among the population at risk remains constant between 2019 and 2040. Note that for countries with anthroponotic VL transmission (i.e. Bangladesh, India, Nepal, Somalia, South Sudan and Sudan), where VL is projected to be eliminated by Malvolti et al. (2021) [14] through existing measures and deployment of a vaccine, we assume that in the absence of vaccine introduction, the population at risk would continue to grow at the 5-year population growth rate from UN/DESA [20].

Similarly, the per person DALY burden of the disease was obtained from the 2019 GBD study for each country and age group considered by dividing the relevant total DALY burden by the incidence, given that the average duration of both CL and VL is less than a year [23]. The 2019 values of the epidemiological parameters used are shown in **Table 2**. This approach was taken due to the lack of country-level data on the per-case DALY burden of the disease. We considered it important to use country-level estimates due to the disparity between countries [2] in terms of clinical and epidemiological presentations, comorbidities, treatment coverage, and fatality rates.

Based on previously developed vaccines [24], efficacy was assumed to be 75% in the primary scenario (based on the efficacy of previously researched leishmanization methods [24,25]). The duration of the efficacy was assumed to be 5 years and an annual discount rate of three percent applied to health gains in the future.

Uncertainty in the above epidemiological variables (incidence and DALYs per person) as well as vaccine efficacy is captured in the estimates by providing lower bound (assuming 50% vaccine efficacy, and lower bound incidence and DALY burden estimates from the 2019 GBD study) and upper bound (assuming 95% vaccine efficacy, and upper bound incidence and DALY burden estimates from the 2019 GBD study) estimates of value-based maximum price.

## Quantity of vaccines demanded

Quantity demanded or demand here refers to the total vaccines projected to be required by a country in a given year based on the target population at risk and rollout constraints, regardless of market price. The vaccine demand projections are based on Malvolti et al. (2021) [14]. This assumed a dual vaccine delivery strategy, including a catch-up campaign at the start followed by rollout in a routine immunization program. Routine immunization includes two age groups—0–4 years, and 5–14 years. The catch-up campaign for CL includes two groups—5–14 years, and 15–29 years; and for VL only the 5–14 years age group was assumed to be targeted. Coverage estimates (those vaccinated as a percentage of those targeted) are based on current vaccines with similar vaccination rollout strategies (see Malvolti et al (2021) [14] for details). Country-wise vaccine demand projections by age are provided in **S1 Table**.

**Table 2. Epidemiological parameters (2019).**

| Country | Visceral Leishmaniasis | | | | | | | Cutaneous Leishmaniasis | | | | | | |
|---|---|---|---|---|---|---|---|---|---|---|---|---|---|---|
| | Population at risk | Incidence (%) among population at risk (0–4 years) | Incidence (%) among population at risk (5–14 years) | Incidence (%) among population at risk (15–19 years) | DALYs lost per person (0–4 years) | DALYs lost per person (5–14 years) | DALYs lost per person (15–19 years) | Population at risk | Incidence (%) among population at risk (0–4 years) | Incidence (%) among population at risk (5–14 years) | Incidence (%) among population at risk (15–19 years) | DALYs lost per person (0–4 years) | DALYs lost per person (5–14 years) | DALYs lost per person (15–19 years) |
| Afghanistan | - | | | | 7.24 [0.02–43.93] | 5.25 [0.02–44] | 7.12 [0.02–44.76] | 11,124,437 | 0.944% [0.32%–1.855%] | 2.129% [0.731%–4.243%] | 1.374% [0.45%–2.797%] | 0.17 [0.29–0.14] | 0.42 [0.76–0.31] | 1.3 [2.53–0.93] |
| Algeria | - | | | | 8.31 [0.03–46.03] | 5.08 [0.02–41.39] | 6.44 [0.03–38.79] | 10,609,819 | 0.187% [0.064%–0.38%] | 0.383% [0.131%–0.785%] | 0.275% [0.094%–0.544%] | 0.03 [0.02–0.04] | 0.03 [0.03–0.04] | 0.03 [0.02–0.04] |
| Bangladesh | 30,955,876 | 0.003% [0.002%–0.005%] | 0.002% [0.001%–0.004%] | 0.001% [0%–0.001%] | 16.32 [0.03–59.42] | 10.4 [0.03–52.98] | 13.77 [0.03–51.75] | - | | | | | | |
| Brazil | 82,117,821 | 0.02% [0.013%–0.03%] | 0.015% [0.01%–0.022%] | 0.005% [0.003%–0.007%] | 14.4 [0.03–39.43] | 10.02 [0.03–36.48] | 14.86 [0.03–42.83] | - | | | | 0.04 [0.03–0.04] | 0.04 [0.04–0.05] | 0.04 [0.04–0.05] |
| China | 232,875,380 | 0.001% [0.001%–0.002%] | 0.001% [0.001%–0.001%] | 0% [0%–0%] | 0.02 [0.01–0.02] | 0.02 [0.01–0.02] | 0.02 [0.01–0.02] | - | | | | 0.03 [0.11–0.03] | 0.04 [0.86–0.04] | 0.09 [7.98–0.05] |
| Ethiopia | 3,424,788 | 0.123% [0.079%–0.184%] | 0.099% [0.063%–0.15%] | 0.035% [0.022%–0.052%] | 23.72 [16.77–28.26] | 17.04 [13.4–18.05] | 17.28 [13.28–18.46] | 5,455,824 | 0.002% [0.001%–0.003%] | 0.003% [0.001%–0.006%] | 0.002% [0.001%–0.004%] | 0.17 [0.24–0.14] | 0.52 [0.76–0.44] | 1.78 [2.66–1.5] |
| Georgia | 2,580,002 | 0.009% [0.005%–0.014%] | 0.007% [0.004%–0.011%] | 0.002% [0.001%–0.004%] | 8.77 [0.03–49.85] | 3.68 [0.02–29.62] | 5.5 [0.02–35.42] | - | | | | 0.03 [0.04–0.04] | 0.03 [0.03–0.03] | 0.03 [0.03–0.03] |
| India | 134,094,347 | 0.017% [0.01%–0.027%] | 0.014% [0.008%–0.022%] | 0.004% [0.002%–0.007%] | 11.92 [0.03–41.88] | 6.93 [0.03–33.3] | 11.57 [0.03–43.9] | 107,275,478 | 0% [0%–0.001%] | 0% [0%–0.001%] | 0% [0%–0.001%] | 0.12 [0.12–0.12] | 0.45 [0.5–0.47] | 1.24 [2.14–0.98] |
| Israel | - | | | | 7.06 [0.03–37.02] | 4.04 [0.02–29.13] | 5.8 [0.02–32.37] | 8,971,638 | 0.002% [0.001%–0.003%] | 0.004% [0.002%–0.007%] | 0.004% [0.002%–0.007%] | 0.03 [0.03–0.04] | 0.03 [0.03–0.04] | 0.03 [0.03–0.04] |
| Kenya | 3,501,646 | 0.075% [0.053%–0.102%] | 0.053% [0.037%–0.072%] | 0.018% [0.012%–0.025%] | 10.48 [7.12–12.59] | 12.26 [8.88–14.61] | 16.09 [11.21–19] | - | | | | 0.11 [0.09–0.12] | 0.22 [0.2–0.25] | 0.46 [0.43–0.5] |
| Morocco | - | | | | 6.44 [0.02–38.77] | 5.31 [0.02–45.06] | 6.72 [0.02–45.06] | 6,364,300 | 0.134% [0.045%–0.277%] | 0.251% [0.084%–0.529%] | 0.149% [0.05%–0.309%] | 0.11 [0.15–0.09] | 0.27 [0.48–0.2] | 0.59 [1.06–0.44] |
| Nepal | 29,942,425 | 0.002% [0.002%–0.004%] | 0.002% [0.001%–0.003%] | 0.001% [0%–0.001%] | 11.3 [0.03–42.47] | 7.7 [0.02–39.25] | 11.76 [0.03–45.9] | - | | | | | | |
| Nigeria | - | | | | 26.32 [24.56–25.13] | 17.98 [16.99–17.41] | 13.73 [12.95–13.36] | 3,238,811 | 0% [0%–0%] | 0% [0%–0.002%] | 0.001% [0%–0.002%] | 0.2 [0.33–0.18] | 0.57 [1.17–0.46] | 1.41 [2.88–1.14] |
| Pakistan | - | | | | 11.75 [0.05–27.37] | 9.17 [0.04–30.07] | 14.55 [0.05–33.61] | 91,841,655 | 0.011% [0.006%–0.018%] | 0.021% [0.011%–0.035%] | 0.017% [0.009%–0.027%] | 0.14 [0.13–0.15] | 0.4 [0.43–0.39] | 1.29 [1.5–1.2] |
| Paraguay | 3,180,239 | 0.004% [0.003%–0.006%] | 0.003% [0.002%–0.005%] | 0.001% [0.001%–0.002%] | 15.33 [0.03–49.23] | 12.35 [0.03–50.69] | 16.4 [0.03–51.78] | | 0.276% [0.094%–0.544%] | 0.911% [0.317%–1.777%] | 0.574% [0.194%–1.137%] | 0.08 [0.13–0.07] | 0.13 [0.14–0.13] | 0.17 [0.18–0.16] |
| Saudi Arabia | - | | | | 8.7 [0.03–46.43] | 4.34 [0.02–28.54] | 5.94 [0.03–31.94] | 3,794,820 | 0.028% [0.012%–0.054%] | 0.208% [0.09%–0.387%] | 0.343% [0.143%–0.646%] | 0.03 [0.03–0.04] | 0.03 [0.02–0.04] | 0.03 [0.01–0.04] |
| Somalia | 2,585,353 | 0.104% [0.066%–0.154%] | 0.08% [0.051%–0.119%] | 0.028% [0.018%–0.041%] | 17.72 [12.7–21.06] | 16.24 [12.48–17.37] | 15.52 [12.16–17.19] | 43,467,592 | 0.015% [0.004%–0.034%] | 0.023% [0.006%–0.051%] | 0.013% [0.003%–0.029%] | 0.03 [0.03–0.03] | 0.03 [0.03–0.03] | 0.03 [0.03–0.04] |
| South Sudan | 2,088,706 | 0.432% [0.28%–0.639%] | 0.345% [0.223%–0.509%] | 0.103% [0.067%–0.157%] | 21.24 [16.28–24.43] | 14.36 [11.5–16.18] | 13.96 [10.32–16.01] | | | | | | | |
| Spain | 37,193,165 | 0.001% [0%–0.001%] | 0% [0%–0.001%] | 0% [0%–0%] | 7.12 [0.03–38.77] | 4.27 [0.03–32.2] | 5.99 [0.03–34.16] | 40,171,954 | 0.004% [0.001%–0.01%] | 0.01% [0.002%–0.023%] | 0.008% [0.002%–0.019%] | 0.03 [0.03–0.03] | 0.03 [0.03–0.03] | 0.03 [0.03–0.03] |
| Sudan | 9,336,300 | 0.084% [0.048%–0.131%] | 0.068% [0.04%–0.108%] | 0.025% [0.014%–0.04%] | 7.37 [0.02–46] | 4.4 [0.02–33.65] | 6.54 [0.02–39.31] | 18,192,904 | 0.184% [0.061%–0.365%] | 0.446% [0.147%–0.898%] | 0.283% [0.091%–0.576%] | 0.12 [0.14–0.11] | 0.26 [0.49–0.19] | 0.78 [1.59–0.55] |
| Syria | - | | | | 8.16 [0.03–52.08] | 6.06 [0.02–52.03] | 6.76 [0.02–45.29] | | | | | 0.08 [0.13–0.07] | 0.12 [0.21–0.1] | 0.16 [0.28–0.12] |
| Tunisia | - | | | | 7.36 [0.03–42.05] | 4.41 [0.02–33.56] | 6.03 [0.03–35.16] | 6,220,910 | | | | 0.03 [0.03–0.04] | 0.03 [0.03–0.04] | 0.03 [0.03–0.04] |
| Turkey | - | | | | 8.97 [0.03–50.43] | 4.96 [0.03–37.88] | 5.86 [0.03–33.13] | | 0.015% [0.004%–0.034%] | 0.023% [0.006%–0.051%] | 0.013% [0.003%–0.029%] | 0.03 [0.03–0.03] | 0.03 [0.03–0.03] | 0.03 [0.03–0.04] |
| Uzbekistan | - | | | | 8.29 [0.02–59.48] | 4.23 [0.02–40.02] | 5.33 [0.02–37.16] | 16,236,757 | 0.017% [0.004%–0.038%] | 0.021% [0.011%–0.048%] | 0.016% [0.003%–0.037%] | 0.03 [0.03–0.04] | 0.03 [0.03–0.04] | 0.03 [0.03–0.04] |

### Health economic analysis–global demand for a leishmaniasis vaccine

We assume that a heath intervention should be provided if it produces more health than could be generated elsewhere in the health care system with the same resources (i.e. the benefits exceed the opportunity costs). For every DALY averted (or QALY gained) from a new intervention, a health system should pay no more than the cost at the margin at which it is already able to avert a DALY from existing interventions (i.e. the marginal productivity; sometimes estimated as a cost-effectiveness threshold (CET)). This approach, previously applied in country-specific studies [26,27], allows us to estimate the maximum ability-to-pay, or the value-based maximum price, for a leishmaniasis vaccine with a given efficacy. A country would demand the required number of courses of the vaccine [14] if the price offered by the manufacturer is below their value-based maximum price, and none if the global market price is above their value-based maximum price. Note that our ability to pay estimates are inclusive of implementation costs incurred for the rollout of the vaccine, i.e. the ability to pay for the medical product itself can be calculated by countries by subtracting their local implementation costs from our estimates.

To determine a price at which a country can afford the hypothetical vaccine requires an estimate of the CET to reflect marginal productivity. We use the 'health budget opportunity cost' approach [28] for CET estimates. A country government may choose to fund the vaccine only if it generates more health than that which would be forgone if its limited health budget is redirected from existing interventions to the vaccine. Country-level CETs have previously been estimated until 2040 by Lomas et al. (2021) [29] based on historical estimates [30] of the marginal productivity of the different countries' health systems. For countries for which these estimates were missing, CETs were projected as an appropriate percentage of the projected GDP per capita [31] based on Ochalek et al. (2020) [32]. Country-level CET estimates used here are provided in **S2 Table**.

In addition to averting DALYs through reduced infections, the vaccine would also reduce system treatment costs which in turn would indirectly increase the ability to pay for the vaccine. The actual reduction in treatment costs for the infected population depends on the expected coverage of treatment, which in most countries would be less than 100%. In the absence of data on leishmaniasis treatment coverage, we project the value-based maximum price under the assumptions of both 0% and 100% treatment coverage to represent its upper and lower bounds.

We assume an average treatment cost per VL case of $541 based on Carvalho et al. (2017) [33]. This estimate includes the average cost through the lifecycle of treatment including pre-diagnosis consultation, drug therapy, hospitalization and ambulatory care until post-treatment consultations. Note that the drug therapy costs are based on the proportion of VL cases treated with meglumine antimoniate, liposomal amphotericin B or amphotericin B deoxycholate respectively in Brazil in 2014. The average treatment cost per CL case is assumed to be $57.6 based on Rodriguez et al. (2019) [34]. This estimate is based on the cost of the drug used (Intralesional pentavalent antimonials (ILPA)) and staff time costs for CL treatment in Bolivia.

Using these concepts, we were able to calculate the value-based maximum price for a course of the leishmaniasis vaccine that each country is able to pay during each year of rollout, given the potential health benefit provided by the vaccine, and the country's CET (Box 1, Eq 1). The demand for vaccines for both CL and VL prevention and the ability-to-pay for each target use case (CL prevention and VL prevention) are aggregated to derive each country's global ability-to-pay for the vaccine (Box 1, Eq 3).

Combined with the vaccine courses estimated to be required for each country, these are used to construct global demand curves for the vaccine during the 11-year period between 2030 and 2040.

## Box 1. Equations to estimate countries' abilities to pay for a leishmaniasis vaccine

The value-based maximum price or ability-to-pay for a course of a leishmaniasis vaccine that each country is able to pay is estimated using the following formula:

$$p_{i,t,\alpha} = \frac{CET_{i,t}\Delta DALY_{i,t,\alpha} + \Delta T_{i,t,\alpha}}{q_{i,t,\alpha}}, \qquad (1)$$

$$\Delta DALY_{i,t,\alpha} = \sum_{\beta}\left[\Delta I_{i,t,\alpha,\beta}\sum_{n=0}^{N}\frac{\Delta DALY\_pp_{i,\alpha,\beta}}{(1+r)^n}\right]$$

$$\Delta T_{i,t,\alpha} = \theta\sum_{n=0}^{N}\frac{\Delta I_{i,t,\alpha}T\_pp_{\alpha}}{(1+r)^n}$$

$$\alpha = \{CL, VL\}$$

*Where i = country*

*t = year of vaccination*

*β = age groups–- 0–4 years, 5–14 years, 15–29 years*

*n = year of vaccine efficacy*

*N = Number of years for which the vaccine is effective*

*r = annual discount rate (%)*

*p = Value-based price for a course of the vaccine (2019 USD)*

*CET = Cost-effectiveness threshold (2019 USD/DALY averted)*

*ΔDALY = Total DALYs averted from the reduction in CL-related/VL-related mortality*

*ΔDALY_pp = Change in DALYs per person infected with CL/VL*

*I = Change in CL/VL incidence as a result of the administration of the vaccine (number of infected people)*

*T = Direct treatment cost of CL/VL (2019 USD)*

*T_pp = Direct treatment cost per case of CL/VL (2019 USD)*

*q = demand for the vaccine (number of vaccine courses)*

*θ = coverage of leishmaniasis (CL and VL) treatment (%)*

To obtain the aggregate demand curve for the period 2030–2040, we obtain the aggregate demand for vaccine courses and the average value-based maximum price for each country across the target use cases as follows:

$$Q_i = \sum_{\alpha}\sum_{t} q_{i,t,\alpha} \qquad (2)$$

$$\bar{p}_i = \frac{\sum_{\alpha}\sum_{t} p_{i,t,\alpha}q_{i,t,\alpha}}{Q_i} \qquad (3)$$

where

$Q_i$ = *country i's aggregate demand for the vaccine between 2030 and 2040 (number of vaccine courses)*

$\overline{p}_i$ = *Average value-based price for a course of the vaccine for country i for the period under consideration (2019 USD)*

For this purpose, we estimate the average value-based maximum price over 11 years, by dividing the sum of the maximum resources which could be committed towards the leishmaniasis vaccine during each year (price times demand) by the aggregate demand between 2030 and 2040.

### Sensitivity analysis

We evaluate the sensitivity of the projected global demand curves to two factors—i) contributions from Gavi, and ii) adjustment for underreporting of leishmaniasis incidence.

Under the first sensitivity analysis, we assess the effect on global demand curves with Gavi contribution towards countries which are expected to be eligible for support between 2030 and 2040 based on GDP per capita projections [31] using Gavi's criterion for support as of 2019 [18]. We assume that a country is eligible for Gavi support during the 11 years under consideration if its projected GDP per capita between 2026 and 2028 is under $1580 (i.e. the country is either in the initial self-financing or preparatory transition phase) or if its GDP per capita has been greater than $1580 for 5 years or less between 2022 and 2028 (i.e. the country is in the accelerated transition phase). Given Gavi's current portfolio of vaccines, we expect Gavi's maximum ability to pay for vaccines to be higher than that of some of the countries eligible for support. Based on previous work on Gavi's willingness to pay for the rotavirus vaccine [35], we assume Gavi's CET to be $285 in 2019 USD. Therefore, under this sensitivity analysis, we re-estimate the demand curve for a leishmaniasis vaccine by increasing the CET value for countries eligible for Gavi support to $285 if their own CET is lower in a given year. Country-level Gavi support projections are provided in **S3 Table**.

Finally, we also assess the potential effect of adjusting for the underreporting of cases on the value-based maximum price, using estimates from Alvar et al. (2012) [36] of CL and VL underreporting by factors in the ranges 3.2–5.7 and 3.5–6.7 respectively (globally).

All the analyses were performed on Excel 2016 and figures were produced in Python 3.8. The workbook and code are publicly available on GitHub (https://github.com/sakshimohan/leish-vaccine).

## Results

### Calculation of value-based maximum price

We calculated the country-wise value-based maximum price per course of the leishmaniasis vaccine and total demand based on Eqs 1,2 and 3, presented in tabular format (**Table 3**) and in the form of a demand curve for the illustrative set of 24 countries (**Fig 1**). As expected, the weighted average of value-based maximum price under the assumption of full coverage of CL and VL treatment is higher (by 19% on average) than under the assumption of no provision of treatment. This is because any treatment expenses saved through reduced incidence increase a country's ability to pay for the vaccine. The average value-based maximum price for the illustrative set of countries, weighted by quantity demanded, is $5.7 [$0.3-$33.7] and $6 [$0.4 - $34.5] under the assumption of 0% and 100% treatment coverage respectively. The intervals around the point estimates

**Table 3. Value-based maximum price for a leishmaniasis vaccine course (2030–2040).**

| Country | Total demand for vaccine courses (2030–2040) | Value-based maximum price per course (assuming treatment coverage = 0%) | Value-based maximum price per course (assuming treatment coverage = 100%) |
|---|---|---|---|
| Afghanistan | 12,963,289 | 3.89 [1.64–7.27] | 7.2 [2.38–15.66] |
| Algeria | 9,167,696 | 2.51 [0.44–7.13] | 3.15 [0.58–8.8] |
| Bangladesh | 11,450,264 | 0.97 [0–7.43] | 1.09 [0.06–7.64] |
| Brazil | 33,953,583 | 61.05 [0.07–363.57] | 61.36 [0.2–364.15] |
| China | 76,399,801 | 0.01 [0–0.02] | 0.03 [0.01–0.05] |
| Ethiopia | 9,532,974 | 12.97 [4.11–27.86] | 13.62 [4.38–29.09] |
| Georgia | 936,404 | 2.15 [0–28.17] | 2.29 [0.06–28.46] |
| India | 135,475,312 | 3.55 [0.01–25.59] | 3.8 [0.14–26.04] |
| Israel | 8,893,083 | 0.25 [0.08–0.6] | 0.26 [0.08–0.62] |
| Kenya | 3,369,092 | 22.41 [7.48–46.62] | 23.56 [8.02–48.63] |
| Morocco | 4,878,092 | 4.47 [1.77–8.78] | 4.87 [1.86–9.84] |
| Nepal | 11,517,485 | 0.78 [0–5.92] | 0.88 [0.05–6.09] |
| Nigeria | 5,015,541 | 0 [0–0.01] | 0 [0–0.02] |
| Pakistan | 101,103,820 | 0.1 [0.04–0.2] | 0.13 [0.05–0.27] |
| Paraguay | 2,078,492 | 15.69 [0.01–107.49] | 15.76 [0.05–107.62] |
| Saudi Arabia | 3,125,075 | 6.21 [0.85–18.57] | 6.64 [0.97–19.59] |
| Somalia | 1,585,209 | 4.65 [2–8.32] | 10.64 [5.47–17.89] |
| South Sudan | 1,049,268 | 61.79 [28.35–111.97] | 85.6 [42.23–149.95] |
| Spain | 10,171,505 | 2.98 [0–42.55] | 2.99 [0.01–42.57] |
| Sudan | 55,713,342 | 2.26 [0.02–26.18] | 2.69 [0.23–26.94] |
| Syria | 21,633,242 | 0.7 [0.28–1.4] | 2.11 [0.61–4.89] |
| Tunisia | 4,447,800 | 1.84 [0.34–5.22] | 2.54 [0.5–7.01] |
| Turkey | 23,864,259 | 0.44 [0.07–1.28] | 0.48 [0.08–1.39] |
| Uzbekistan | 15,730,233 | 0.05 [0.01–0.15] | 0.08 [0.01–0.26] |
| **Weighted average** | | 5.67 [0.27–33.74] | 6.04 [0.42–34.51] |

represent the lower bounds (assuming 50% vaccine efficacy and lower bound epidemiological indicators) and upper bounds (assuming 95% vaccine efficacy and upper bound epidemiological indicators) of the weighted average of value-based maximum price.

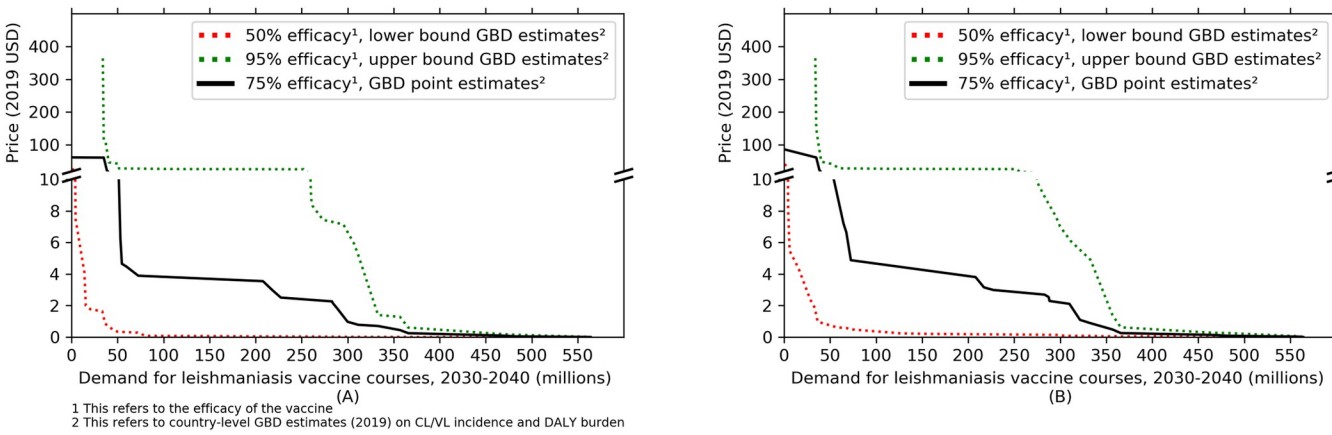

**Fig 1.** Illustrative global demand curve for a leishmaniasis vaccine between 2030 and 2040: (A) assuming treatment coverage = 0%, (B) assuming treatment coverage = 100%.

### Sensitivity analysis

**Impact of Gavi support for vaccine introduction.** The results described above treat countries as independent buyers of the vaccine whose ability to pay per vaccine course depends on their respective CETs. However, international donors are often able to ensure the expansion of important health interventions to low and lower-middle income countries even when these may be locally cost-ineffective as a result of budget constraints. We consider the effect of future Gavi funding of a potential leishmaniasis vaccine for countries eligible for its support based on current criteria [18]. We project that 11 of the 24 countries in our illustrative list will be in one of the Gavi support phases (S3 Table), of which six countries have a CET lower than $285 in 2030. Using a CET of $285/DALY averted for these six countries, provides an alternate demand curve (Fig 2). The weighted mean value-based maximum price increases by 12% under both treatment coverage scenarios (Table 4).

### Sensitivity to underreporting

The final sensitivity analysis adjusting for underreporting increases the average ability to pay to $19.6 [$0.9-$117]—$20.9 [$1.4-$119.6] (an increase of approximately 250%) under the assumption of underreporting by a factor of 3.2 and 3.5 for CL and VL, respectively. These figures increase to $37.6 [$1.7–224.8]—$39.9 [$2.6-$230] (an increase of approximately 560%) when the upper bound underreporting factors of 5.7 and 6.7 are applied for CL and VL respectively (Fig 3 and Table 4).

## Discussion

This study has sought to provide a generalizable approach to estimating the commercial and public health value of new technologies in development relying primarily on publicly available GBD data. This simplified approach is easily replicable and can be used to guide discussions and investments into health technology development, particularly in low and middle-income countries (LMICs), which face significant constraints in acquiring and generating evidence compared with higher-income countries.

The utility of this approach is demonstrated by projecting the economic feasibility of a leishmaniasis vaccine based on currently available estimates of CETs based on marginal productivity, disease incidence and burden of disease. While other studies have previously tried to

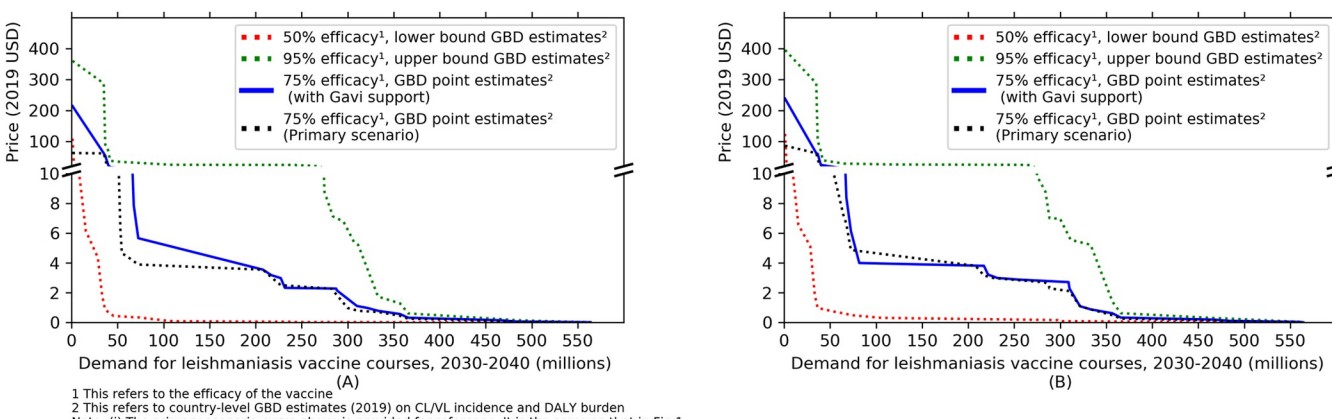

**Fig 2.** Illustrative global demand curve for a leishmaniasis vaccine between 2030 and 2040 including Gavi support: (A) assuming treatment coverage = 0%, (B) assuming treatment coverage = 100%.

**Table 4. Value-based maximum price for a leishmaniasis vaccine course (2030–2040)–Sensitivity analyses.**

| Country | With Gavi support | | Underreporting by a factor of 3.2 (CL) and 3.5 (VL) | | Underreporting by a factor of 5.7 (CL) and 6.7 (VL) | |
|---|---|---|---|---|---|---|
| | VBP (assuming treatment coverage = 0%) | VBP (assuming treatment coverage = 100%) | VBP (assuming treatment coverage = 0%) | VBP (assuming treatment coverage = 100%) | VBP (assuming treatment coverage = 0%) | VBP (assuming treatment coverage = 100%) |
| Afghanistan | 13.18 [4.38–19.44] | 17.38 [5.12–27.83] | 12.56 [5.28–23.46] | 23.25 [7.69–50.56] | 22.06 [9.27–41.2] | 40.83 [13.5–88.78] |
| Algeria | 3.18 [0.44–7.13] | 3.99 [0.58–8.8] | 8.09 [1.41–23.02] | 10.18 [1.88–28.4] | 14.21 [2.47–40.43] | 17.88 [3.31–49.87] |
| Bangladesh | 0.97 [0–6.79] | 1.09 [0.07–6.97] | 3.38 [0–25.8] | 3.79 [0.21–26.52] | 6.5 [0.01–49.65] | 7.29 [0.41–51.04] |
| Brazil | 61.05 [0.1–287.03] | 61.36 [0.31–287.49] | 212 [0.23–1262.53] | 213.09 [0.71–1264.56] | 407.95 [0.44–2429.52] | 410.06 [1.36–2433.44] |
| China | 0.01 [0.01–0.02] | 0.03 [0.02–0.04] | 0.03 [0.01–0.07] | 0.1 [0.05–0.19] | 0.07 [0.03–0.13] | 0.2 [0.09–0.36] |
| Ethiopia | 12.98 [6.16–22.01] | 13.62 [6.57–22.98] | 45.04 [14.27–96.74] | 47.28 [15.22–101] | 86.65 [27.46–186.14] | 90.96 [29.29–194.34] |
| Georgia | 2.15 [0.01–22.24] | 2.29 [0.08–22.47] | 7.46 [0.01–97.83] | 7.96 [0.2–98.83] | 14.35 [0.02–188.26] | 15.31 [0.38–190.17] |
| India | 3.55 [0.01–23.46] | 3.81 [0.16–23.87] | 12.31 [0.02–88.85] | 13.21 [0.47–90.43] | 23.69 [0.04–170.97] | 25.42 [0.9–174.02] |
| Israel | 0.32 [0.08–0.6] | 0.32 [0.08–0.62] | 0.8 [0.25–1.95] | 0.83 [0.25–2] | 1.41 [0.43–3.43] | 1.45 [0.45–3.52] |
| Kenya | 22.41 [11.22–36.81] | 23.56 [12.04–38.39] | 77.83 [25.97–161.89] | 81.83 [27.87–168.86] | 149.76 [49.98–311.54] | 157.47 [53.63–324.95] |
| Morocco | 5.67 [1.77–8.78] | 6.17 [1.86–9.84] | 14.44 [5.7–28.35] | 15.73 [5.99–31.77] | 25.36 [10.02–49.78] | 27.63 [10.52–55.8] |
| Nepal | 0.78 [0–5.4] | 0.88 [0.06–5.56] | 2.71 [0–20.57] | 3.06 [0.19–21.16] | 5.22 [0.01–39.58] | 5.88 [0.36–40.71] |
| Nigeria | 0.01 [0–0.01] | 0.01 [0–0.02] | 0.01 [0–0.04] | 0.02 [0–0.05] | 0.02 [0–0.07] | 0.03 [0.01–0.09] |
| Pakistan | 0.14 [0.04–0.22] | 0.19 [0.06–0.3] | 0.32 [0.12–0.63] | 0.44 [0.17–0.88] | 0.56 [0.22–1.11] | 0.77 [0.29–1.54] |
| Paraguay | 15.69 [0.02–84.86] | 15.76 [0.07–84.96] | 54.49 [0.05–373.27] | 54.73 [0.16–373.72] | 104.85 [0.1–718.29] | 105.33 [0.31–719.16] |
| Saudi Arabia | 7.86 [0.85–18.57] | 8.41 [0.97–19.59] | 20.04 [2.75–59.95] | 21.43 [3.15–63.24] | 35.2 [4.84–105.28] | 37.63 [5.53–111.06] |
| Somalia | 53.29 [25.3–88.88] | 59.29 [29.12–97.78] | 16.13 [6.95–28.88] | 36.96 [18.98–62.12] | 31.05 [13.38–55.58] | 71.12 [36.53–119.53] |
| South Sudan | 213.97 [108.61–360.13] | 237.78 [124.02–395.33] | 214.57 [98.44–388.83] | 297.26 [146.65–520.73] | 412.9 [189.43–748.23] | 572.03 [282.21–1002.06] |
| Spain | 2.98 [0.01–33.59] | 2.99 [0.01–33.61] | 10.35 [0.02–147.76] | 10.38 [0.03–147.83] | 19.92 [0.03–284.34] | 19.97 [0.05–284.47] |
| Sudan | 2.28 [0.02–24.1] | 2.71 [0.26–24.8] | 7.85 [0.06–90.89] | 9.34 [0.81–93.5] | 15.09 [0.11–174.86] | 17.94 [1.55–179.85] |
| Syria | 1.11 [0.35–1.74] | 2.89 [0.67–5.22] | 2.27 [0.91–4.52] | 6.81 [1.96–15.77] | 3.99 [1.6–7.94] | 11.95 [3.44–27.7] |
| Tunisia | 2.33 [0.34–5.22] | 3.22 [0.5–7.01] | 5.93 [1.1–16.84] | 8.2 [1.6–22.62] | 10.41 [1.94–29.57] | 14.4 [2.81–39.73] |
| Turkey | 0.56 [0.07–1.28] | 0.61 [0.08–1.39] | 1.43 [0.23–4.14] | 1.55 [0.25–4.48] | 2.52 [0.4–7.27] | 2.73 [0.44–7.87] |
| Uzbekistan | 0.06 [0.01–0.15] | 0.11 [0.01–0.26] | 0.15 [0.02–0.47] | 0.27 [0.04–0.83] | 0.27 [0.03–0.83] | 0.48 [0.06–1.46] |
| **Weighted average** | 6.37 [0.61–28.97] | 6.78 [0.78–29.69] | 19.62 [0.92–117.01] | 20.87 [1.41–119.57] | 37.59 [1.71–224.8] | 39.91 [2.64–229.51] |

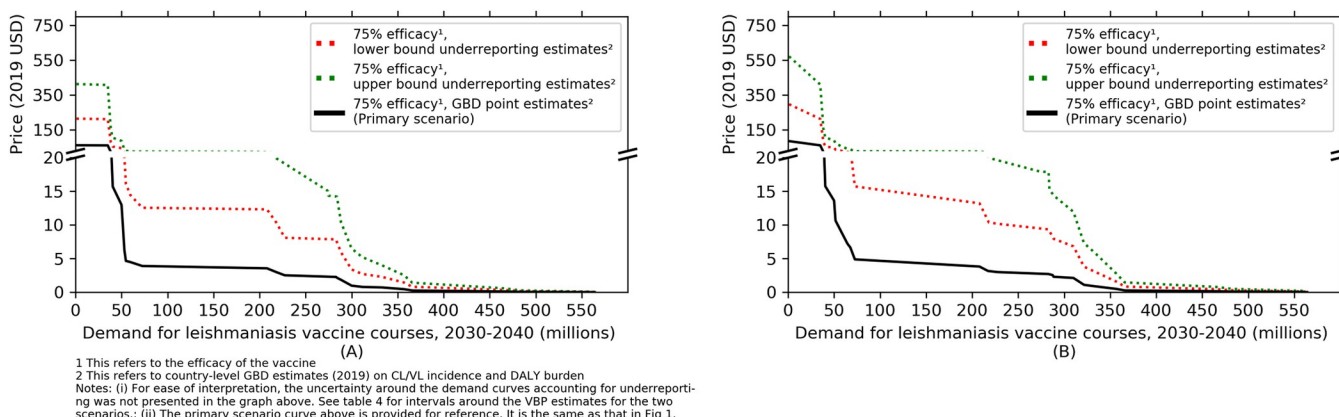

1 This refers to the efficacy of the vaccine
2 This refers to country-level GBD estimates (2019) on CL/VL incidence and DALY burden
Notes: (i) For ease of interpretation, the uncertainty around the demand curves accounting for underreporting was not presented in the graph above. See table 4 for intervals around the VBP estimates for the two scenarios.; (ii) The primary scenario curve above is provided for reference. It is the same as that in Fig 1.

**Fig 3.** Sensitivity of value-based maximum price to underreporting: (A) assuming treatment coverage = 0%, (B) assuming treatment coverage = 100%.

 

estimate the cost-effectiveness of vaccines [37,38] and the monetary value of health technologies [26,27], our approach is novel for its global focus and simplicity as well as the incorporation of practical considerations including a realistic timescale of when the product is expected to be available for distribution, gradual rollout and an evolving expected marginal productivity of health systems.

Our results demonstrate that both the quantity of vaccines estimated to be required by the countries considered, which represent a majority of the global burden of disease from leishmaniasis, as well as their ability-to-pay make the vaccine commercially attractive to potential manufacturers. The global demand stands at over 560 million courses, and the value-based maximum price per course, given the current estimates of incidence and population at risk, is higher than $5 for nearly a third of the 24 countries considered (with a weighted average of $5.7 - $6 in the primary scenario). Assuming a full course of two doses and an expected manufacturing cost of $2–3 per dose, based on adenovirus vaccines [39] similar to ChAd63-KH (the only leishmaniasis vaccine currently recruiting into clinical trial [12]), a leishmaniasis vaccine of this type would be commercially viable. The wide range of value-based maximum prices across different countries also presents an opportunity for differential pricing to secure wide access. With possible future contributions from Gavi considering its current willingness to pay for the rotavirus vaccine [35], we estimate that the global demand curve would move further upwards. A similar upward effect in abilities to pay is observed with adjustment for underreporting.

It should be noted that the prices presented above represent the maximum full health system cost per vaccinated individual that countries can afford in the future. In other words, in order to determine the value-based maximum price for the vaccine itself, countries will also need to consider the number of doses required per course as well as the implementation costs. We have not included implementation costs in our calculations because of the vast uncertainty in these costs and variability across settings [40]. For instance, the choice of vaccine rollout strategy (such as combining it with other immunization programs) would result in a significant difference in the unit cost of implementation. These costs could also make the vaccine unaffordable for some countries. Furthermore, we had to make several simplifying assumptions due to evidence and data gaps as well as to ensure that our results remained interpretable. The absence of context-specific infectious disease models available for all the countries meant that we were unable to capture the effect of disease dynamics and interactions of a potential vaccine with other disease control and management interventions (such as vector control), which could increase or decrease the value of the vaccine for a country. In the absence of epidemiological projections for leishmaniasis, we had to assume that the incidence of disease would remain constant between 2019 and 2040, if no vaccine were to become available. Finally, the quality of our results depends on the quality of the underlying data on disease demographics, burden of disease, and vaccine rollout projections, which can only be addressed through better country level data; for instance, the wide confidence intervals for CL and VL incidence and disease burden from 2019 GBD estimates lead to a large amount of uncertainty in our estimates of countries' ability to pay.

However, by being conservative in our assumptions, we believe that overall our projections underestimate the ability to pay for a leishmaniasis vaccine for a range of reasons including the exclusion of post-kala-azar dermal leishmaniasis (PKDL) and its effect on VL transmission [41], exclusion of disease dynamics or transmission effects, exclusion of psychosocial and mental health effects of the disease (which could amount to six times the current estimate of DALY burden for CL [4]), and exclusion of treatment cost for leishmaniasis-HIV coinfection (which would increase the treatment cost per VL case by up to four times [33]). Updating our assumptions based on a combination of all these factors could increase our estimates of maximum

ability to pay. However, there are several other sources of uncertainty, imposed by a continuously evolving health sector landscape, which can only be addressed by updating these estimates as and when updated information becomes available. Therefore, our demand and price projections are far from definite but shine a light on important data gaps and uncertainties in characterizing the leishmaniasis epidemic, addressing which will be crucial to better understanding the future value of a vaccine against these diseases.

With better data, a full epidemiological model capturing disease dynamics should form the basis of projections of the public health value of potential technologies. Such analysis is rarely feasible before a product enters a market due to lack of resources and analytical capacity, as well as global data on necessary parameters. Our framework overcomes these challenges, albeit through various simplifications, and we suggest that our results can be used to guide investments into improving the data available on leishmaniasis. In addition, our results should help set in motion global discussions on the public health value and commitment towards a leishmaniasis vaccine and help direct vaccine target product profiles to ensure economic feasibility.

## Supporting information

**S1 Table. Vaccine rollout projection (2030–2040).**
(DOCX)

**S2 Table. Projected Cost-effectiveness Thresholds (CETs) (2030–2040, 2019 USD).**
(DOCX)

**S3 Table. Projected GAVI support status in 2030.**
(DOCX)

## Author Contributions

**Conceptualization:** Sakshi Mohan, Paul Revill, Stefano Malvolti, Melissa Malhame, Mark Sculpher, Paul M. Kaye.

**Data curation:** Sakshi Mohan.

**Formal analysis:** Sakshi Mohan.

**Funding acquisition:** Paul Revill, Paul M. Kaye.

**Investigation:** Sakshi Mohan, Stefano Malvolti.

**Methodology:** Sakshi Mohan, Paul Revill, Stefano Malvolti, Melissa Malhame, Mark Sculpher, Paul M. Kaye.

**Supervision:** Paul Revill, Paul M. Kaye.

**Writing – original draft:** Sakshi Mohan, Paul Revill.

**Writing – review & editing:** Stefano Malvolti, Melissa Malhame, Mark Sculpher, Paul M. Kaye.

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
