## [Decision Letter · Decision Letter 0]

20 Dec 2021

Dear Mohan,

Thank you very much for submitting your manuscript "Estimating the global demand curve for a leishmaniasis vaccine: a generalisable approach based on global burden of disease estimates" for consideration at PLOS Neglected Tropical Diseases. As with all papers reviewed by the journal, your manuscript was reviewed by members of the editorial board and by several independent reviewers. In light of the reviews (below this email), we would like to invite the resubmission of a significantly-revised version that takes into account the reviewers' comments. 

If authors decide to submit a revised version, please make sure to address each comment and pay particular attention to comments on model validity, uncertainty, and sensitivity analysis.

We cannot make any decision about publication until we have seen the revised manuscript and your response to the reviewers' comments. Your revised manuscript is also likely to be sent to reviewers for further evaluation.

Sincerely,

Claudia Munoz-Zanzi

Associate Editor

Helen Price

Deputy Editor

If authors decide to submit a revised version, please make sure to address each comment and pay particular attention to comments on model validity, uncertainty, and sensitivity analysis.

Reviewer's Responses to Questions

**Key Review Criteria Required for Acceptance?**

**Methods**

-Are the objectives of the study clearly articulated with a clear testable hypothesis stated?

-Is the study design appropriate to address the stated objectives?

-Is the population clearly described and appropriate for the hypothesis being tested?

-Is the sample size sufficient to ensure adequate power to address the hypothesis being tested?

-Were correct statistical analysis used to support conclusions?

-Are there concerns about ethical or regulatory requirements being met?

Reviewer #1: Methods and underlying assumptions and sensitivity analyses are well described. This is a generalised approach that recognises data limitations but, as such, does provide a foundational piece of work that can be adapted for diseases other than VL/CL. The sample size of selected countries based on overall VL/CL burden seems appropriate.Overall developmental need for the analysis is well described and the work seeks top bridge information gaps between demand, supply and possible value price points for new products (in this case a vaccine for VL/CL).

Reviewer #2: --SPECIFIC COMMENTS 

As for the model itselfI would encourage the authors to adopt a probabilistic approach in which all 14 input factors are characterized by their probability distribution function (pdfs) and the model is run in a Monte Carlo approach (see Saltelli A (2008)). Yet, model output can be characterized probabilistically by accounting for the full variability and uncertainty of values. If averages are provided I would feel much more comfortable 

In any event, overall it would be nice to see some pdfs of epitomic countries and maps of countries color-coded based on incidence under different scenarios. I believe tables are hard to read and communicate little. They can be placed as SI. 

If possible I believe it would better to present results in a portfolio approach, that is total incidence reduction vs. resources needed or value-based maximum price, rather than price as a function of demand. I believe a population outcome should be the main evaluative outcome and not the value-based maximum price. Efficiency frontiers can be determined and displayed as a set of points and each point has different efficiency. it is hard to imagine that efficiency and treatment coverage is constant for so many solutions determined by demand and price (although theoretically possible). 

The probabilistic approach is also very useful for a non-linear sensitivity analysis that can be carried out even by using a variance-based approach but considering all input factors as changing together (see Pianosi et al (2016)). At the moment the authors used only a simple linear sensitivity approach on two parameters to alter vs. 14 input factors, as well as the neglected non-linear interactions to define which factor is truly altering model outputs. Of course later on you can analyze model output dependent on efficiency and treatment coverage but a-priori you cannot know whether these factors are the most important one (additionally these factors are non-linearly dependent with each other).

Lastly, I am not sure why the axes of the graph is cut ...

--REFERENCES 

Chan LYH et al (2021)

COVID-19 non-pharmaceutical intervention portfolio effectiveness and risk communication predominance

Scientific Reports volume 11, Article number: 10605 (2021)

Servadio J et al., (2020)

https://journals.plos.org/plosone/article?id=10.1371/journal.pone.0235920

Information differences across spatial resolutions and scales for disease surveillance and analysis: The case of Visceral Leishmaniasis in Brazil

Pianosi F et al. (2016)

Sensitivity analysis of environmental models: A systematic review with practical workflow

Environmental Modelling & Software

Volume 79, May 2016, Pages 214-232

- https://www.safetoolbox.info/info-and-documentation/

Saltelli A (2008)

Global Sensitivity Analysis. The Primer

Reviewer #3: - The objective of providing a simple framework for estimating vaccine affordability and testing it in a range of countries for Leishmaniasis is clearly stated.

- There is no test of the validity of the estimates which is especially important as the authors suggest using the methodology as a general framework. The authors state that more complex models exist for some countries for their use-case but there is no comparison done of the estimates from the simplified framework to more complex ones making an assessment of the quality and validity difficult.

- The authors do not include uncertainty in the used parameters but provide point estimates without CIs. As there is substantial uncertainty in a lot of the used data especially from GBD it is absolutely necessary to include those. Point estimates alone are uninformative.

- The results are strongly dependent on the chosen efficacy for the vaccine which is clear from both the formulas and the results. As such I don't consider varying the efficacy a sensitivity analysis as we know it's very sensitive to it but rather complete results should be shown for a range of values.

Reviewer #4: Yes to all of the above; no concerns.

**Results**

-Does the analysis presented match the analysis plan?

-Are the results clearly and completely presented?

-Are the figures (Tables, Images) of sufficient quality for clarity?

Reviewer #1: Yes

Reviewer #2: Results should also consider non-linearity in model factors (for sensitivity analysis)

Reviewer #3: - The result section is very short and numbers are given with too many digits in the table which gives an impression of certainty that is not warranted. All results need CIs as mentioned in previous comments.

Reviewer #4: Yes to the first two items. The figures I received were a bit low-resolution and showed artefacts (horizonal smears) at the breakpoints; this has been noted in the request for minor revisions.

**Conclusions**

-Are the conclusions supported by the data presented?

-Are the limitations of analysis clearly described?

-Do the authors discuss how these data can be helpful to advance our understanding of the topic under study?

-Is public health relevance addressed?

Reviewer #1: Yes. Limitations in terms of data are described. Underlying assumptions are described with underpinning rationale.

Reviewer #2: --GENERAL COMMENTS

The manuscript ''Estimating the global demand curve for a leishmaniasis vaccine: a generalisable approach based on global burden of disease estimates'' is interesting and fit the journal. While I do not have major concerns about the topic or the results (if I just focus on the correctness of what has been done) I question some conceptual arguments and the sensitivity analysis of the model. I believe theoretically the paper has its own validity but not rather practically since it neglects many factors such as multiple interventions, countries and diseases, whose synergies with leishmaniasis is fundamental for assessing the demand and effectiveness curve for a leishmaniasis vaccine. 

I do like the incidence-economic evaluation approach however, I think the authors should discuss the limitations of their study that as it is a modeling exercise only without the consideration of other factors to make it applicable. 

Whenever an action is evaluated, there should be the recognition of other potential interventions that aim to decrease disease incidence (see Chan LYH et al (2021) in a portfolio approach; the case study was done for COVID and NPI but it can be applied to any disease and intervention). For leishmaniasis other factors are possible such as ecosystem management, targeting either habitats (ecohydrological controls) and specific species as animal hosts (Servadio J et al., (2020)), leaving aside education campaigns. 

Secondly, at the world and country scale, other diseases should be considered when prioritizing a vaccine. This is the third element that is neglected and then should be discussed.

Lastly, country interdependencies is extremely important when evaluating the effectiveness of a vaccine because boundaries are less and less important in disease transmission but also in terms of vaccine development decision (at the world scale). 

Certainly the consideration of all these factors is hard but doable. It is ok not to include these features in this paper BUT I believe these three elements must be discussed because a vaccine cannot just be evaluated by considering leishmaniasis by itself country by country.

Reviewer #3: - The authors state that their framework overcomes the challenges of a lack of detailed data for a full epidemiological model by making simplifying assumptions. There is no clear discussion of what those assumptions are and how they might affect the estimates.

- Also a discussion of how uncertainty in the data that is used even for the simplified framework might affect the usefulness of the estimates is needed. Currently this topic seems to have been avoided in the manuscript.

Reviewer #4: Yes to all, no concerns.

**Editorial and Data Presentation Modifications?**

Reviewer #1: (No Response)

Reviewer #2: Data should be better presented as highlited in my review of results

Reviewer #3: (No Response)

Reviewer #4: General: You do not state the software in which this analysis was conducted, nor do you describe the public availability of code that would make this analysis truly replicable. Please list software names and versions, and either link to a public code repository, include your code in the supplementary materials, or explain why you are not able to share your code.

Text:

- grammatical note: please leave a space between words and citations.

- a brief description of what sort of disease leishmaniasis is (what are the vectors, etc) would strengthen the introduction's discussion of different control options.

- lines 88-94: can you cite literature to support the importance of these two stakeholder groups in vaccine development?

- line 111: "is" endemic, not "in" endemic

- lines 129-131: state why the countries representing the other 20% of burden were excluded.

- use either "Gavi" or "GAVI" throughout the document.

- In the discussion, please also note the limitation that GBD estimates themselves are modeled and highly uncertain, especially in countries with little or no data.

- If the COVID-19 pandemic has any potential impact on the results or implications of this analysis, please state them. 

Figures and tables, etc:

-Box 1: please include units for all definitions.

-Figures: In the versions of the images I received, there are strange artefacts (smeared horizontal lines) at the breakpoints of the plots. Please ensure these are removed prior to resubmission.

-Figures 2-4 would be more effective if you also included the comparison curve from Figure 1 on the same plot-- it is difficult to compare otherwise.

A suggestion, but not a requirement: Your figures show demand curves, but your discussion and results focus primarily on value-based maximum price. A plot of value-based maximum price across your different tested scenarios would be extremely useful as a main outcome figure.

**Summary and General Comments**

Reviewer #1: Overall I found this a helpful analysis, recognising information and data gaps but clearly articulating the need to bridge these and to provide an approach to estimating demand and maximum value pricing against a set of assumptions which were well described and justified. Sensitivity analyses help provide further detail on upper and lower bounds of confidence. The paper uses “ability to pay” in many places with regard to domestic payers and later in the text when referring to GAVI it speaks of willingness to pay. This might be worth further exploration. We know that domestic financing decisions are complex and not totally determined by supply/demand and maximum value pricing but also based on political considerations and there for overall willingness to pay based on competing opportunities. This also speaks to the overall conclusion of the paper, with which I do not disagree.

Reviewer #2: --RECOMMENDATION

I recommend Major Revisions considering my above comments. The paper is interesting but needs to be improved theoretically by listing limitations to real application and a better presentation of results is needed: the latter by jopefully considering the probabilistic distribution of model factors and their interdependencies. I also highly suggest to make use of maps and compare ''extreme'' / epitomic countries against each other.

Reviewer #3: The authors present a framework to calculate the commercial and public health value of a vaccine based on limited data which is generally available for a wide range of countries and diseases. This is clearly an interesting and worthwile objective, however, there are some further steps needed to be able to assess the utility of the model. Uncertainty in model parameters used needs to be included to generate valid CIs for the model estimates, the model needs to be validated against more complex models in settings where those exist, and a clear discussion of the simplifying assumptions behind the model is needed.

Reviewer #4: This analysis is clear, well-written, and valuable to the NTD community. To my knowledge, it is novel to the field. While it thoroughly and effectively presents both methods and results, I have included some suggestions to increase clarity and rigor. In particular, a statement about the public availability of code is important from a replicability perspective. I have also made some suggestions to improve figure quality and clarity. I have no other major concerns.

PLOS authors have the option to publish the peer review history of their article (what does this mean?). If published, this will include your full peer review and any attached files.

Reviewer #1: Yes: Simon Bland CBE

Reviewer #2: No

Reviewer #3: No

Reviewer #4: No
---

## [Decision Letter · Decision Letter 1]

5 May 2022

Dear Mohan,

We are pleased to inform you that your manuscript 'Estimating the global demand curve for a leishmaniasis vaccine: a generalisable approach based on global burden of disease estimates' has been provisionally accepted for publication in PLOS Neglected Tropical Diseases.

Best regards,

Helen P Price, PhD

Deputy Editor

Helen Price

Deputy Editor

The reviewers are satisfied that the comments have been addressed and the manuscript is now suitable for publication.

Reviewer's Responses to Questions

**Key Review Criteria Required for Acceptance?**

**Methods**

-Are the objectives of the study clearly articulated with a clear testable hypothesis stated?

-Is the study design appropriate to address the stated objectives?

-Is the population clearly described and appropriate for the hypothesis being tested?

-Is the sample size sufficient to ensure adequate power to address the hypothesis being tested?

-Were correct statistical analysis used to support conclusions?

-Are there concerns about ethical or regulatory requirements being met?

Reviewer #2: The paper can be accepted for publication since it answered review's comments satisfactorily

Reviewer #4: yes to all; no concerns.

**Results**

-Does the analysis presented match the analysis plan?

-Are the results clearly and completely presented?

-Are the figures (Tables, Images) of sufficient quality for clarity?

Reviewer #2: v

Reviewer #4: My previous concerns about the figures have been rectified. I would recommend clarifying in the caption to figure 3 that the mean estimate is not within the upper/lower confidence bounds because it is not capturing any underreporting-- this was clear as I read through the paper, but not when I looked at the figures in isolation.

**Conclusions**

-Are the conclusions supported by the data presented?

-Are the limitations of analysis clearly described?

-Do the authors discuss how these data can be helpful to advance our understanding of the topic under study?

-Is public health relevance addressed?

Reviewer #2: The paper can be accepted for publication since it answered review's comments satisfactorily

Reviewer #4: Yes to all; no concerns.

**Editorial and Data Presentation Modifications?**

Reviewer #2: The paper can be accepted for publication since it answered review's comments satisfactorily

Reviewer #4: Lines 89-90 you reference an occurrence in 2021 as forthcoming-- consider changing tense.

Line 212: I believe you intended a comma rather than a period.

Line 273: "in", not "on", Excel and Python, and please capitalize Excel and specify what versions of the software you used. It is particularly important to note whether you used Python 2 or 3.

**Summary and General Comments**

Reviewer #2: The paper can be accepted for publication since it answered review's comments satisfactorily

Reviewer #4: I think this is a well-thought-out paper and I commend the authors for their work and revisions.

PLOS authors have the option to publish the peer review history of their article (what does this mean?). If published, this will include your full peer review and any attached files.

Reviewer #2: No

Reviewer #4: No

---

## [Editor Report · Acceptance letter]

8 Jun 2022

Dear Mohan,

We are delighted to inform you that your manuscript, "Estimating the global demand curve for a leishmaniasis vaccine: a generalisable approach based on global burden of disease estimates," has been formally accepted for publication in PLOS Neglected Tropical Diseases.

Best regards,

Shaden Kamhawi

co-Editor-in-Chief

Paul Brindley

co-Editor-in-Chief
